# A study exploring procedures used to select and analyse microenterprises for persons with disabilities

Luther Lebogang Monareng[1,2]*, Shaheed Mogammad Soeker[3], Deshini Naidoo[1]

1 Department of Occupational Therapy, College of Health Sciences, University of KwaZulu-Natal, Durban, South Africa, 2 University College London, London, United Kingdom, 3 Occupational Therapy Department, University of the Western Cape, Cape Town, South Africa

* leboganglolo@gmail.com, monarengl@ukzn.ac.za

## Abstract

### Background

Attaining the United Nations' 2030 Sustainable Goals, such as fighting poverty and involving adults in work, requires the involvement of professionals such as occupational therapists. Persons with disabilities are among the adults to whom occupational therapists provide work or vocational rehabilitation services for productivity and well-being. Occupational therapists have skills such as analysing tasks to determine associated demands and requirements. The analysis ensures task feasibility, suitability, and matching of persons with disabilities. However, anecdotal evidence indicates that occupational therapists lack a systematic and practical approach to select and analyse suitable microenterprises for individuals with disabilities they serve. The objectives of this study were to address the gap in occupational therapy by exploring the i) selection of a suitable microenterprise for persons with disabilities and ii) factors to consider when analysing microenterprises to ensure successful outcomes. Thus, this study aims to explore procedures used to select and analyse microenterprises for persons with disabilities.

### Methods

Seventeen participants participated in this exploratory qualitative research, three male. They were occupational therapists based in academia, clinical settings and learners with Special Education Needs Schools, respectively. Purposive and snowball sampling were used to recruit the participants. Data was analysed using thematic analysis using the hybrid inductive and deductive approach. Ethical clearance was issued by the University of KwaZulu-Natal's Biomedical Research Ethics Committee.

**Data availability statement:** All raw data files are available from Open Science Framework (OSF). The full reference is as follows: Monareng LL. PhD Raw Data. Data collection from 2023 to 2024 [Internet]. OSF; 2025. Available from: https://osf.io/pdfks/.

**Funding:** This research was supported by i) South Africa's National Research Foundation (NRF), ii) University Capacity Development Grant Funding (UCDP) from the University of KwaZulu Natal and iii) The University of KwaZulu-Natal's Step Up Programme, Department of Higher Education and Training (DHET) and Newton Fund. The funders had no role in study design, data collection and analysis, decision to publish, or preparation of the manuscript.

**Competing interests:** The authors have declared that no competing interests exist.

## Results

Two themes emerged. Theme one: The process of selecting an appropriate micro-enterprise. This theme describes the participant's insight into the approach to micro-enterprise selection, the use of the microenterprise list or options available, and the distinct features of these microenterprises. Theme two: Factors to consider when analysing a suitable microenterprise as a placement option. The theme explores microenterprise accessibility, the key role players involved, business demands, and funding availability and access.

## Conclusion/discussion/interpretation

For microenterprise selection and suitability analysis, a comprehensive, systematic and contextualised approach is crucial to facilitate self-employment as a viable career choice. Such includes integrating a list of microenterprises found in South Africa for career choice consideration. Moreover, there are essential factors for consideration, including the involvement of various key role players, legal and regulatory frameworks, funding sources and leveraging the strengths of persons with disabilities. These factors, effectively integrated with occupational therapists' expertise in vocational rehabilitation, can enhance the vocational success of persons with disabilities.

## 1. Introduction

The introductory section unpacks the dynamics surrounding the selection of suitable microenterprises in self-employment and the factors to consider when evaluating this type of employment as a viable placement opportunity. This section provides an overview, research on self-employment, examination of impact, the key role players involved and how they assess and implement intervention. The last part details the aims and objectives. This structured approach underscores the imperative of a systematic and evidence-based approach.

### 1.1. Overview

Key role players should develop a comprehensive and contextual understanding of self-employment to support individuals involved in microenterprises. A thorough interrogation of the approach, types, and a comprehensive list of available microenterprises and necessary resources is crucial for establishing and sustaining these ventures. These microenterprises are either face-to-face or online-based and not limited to specific regions, i.e., they are found globally and transcend geographical demarcations [1–4]. For instance, microenterprises are found in Africa [2,5–11], Asia [1], Australia [3,12] and the United States [4,13–17]. Despite variations in context and resources, commonalities exist among microenterprises worldwide, including similarities in types, challenges, and benefits as highlighted below. The pervasiveness of microenterprises suggests these opportunities warrant exploring. To effectively leverage these opportunities, it is crucial to consider various factors, including customer

relations and contextual and geographical resources [18]. By understanding these elements, microenterprises can be tailored to meet the needs of persons with disabilities, ensuring a more effective selection and matching process. Empirical research is essential to inform this process and ensure that strategies used are evidence-based.

## 1.2. Research on self-employment

The theoretical foundation for self-employment through microenterprises for persons with disabilities is increasingly recognised, yet empirical research remains critically underdeveloped. This is supported by researchers such as Monareng et al. (2023) [19] who highlight the need to bridge the existing research gap in self-employment. This study aligns with global bodies' initiatives, such as the United Nations' 2030 Sustainable Development Goals, which aim to mitigate poverty and foster meaningful employment for all adults [20]. Moreover, other research findings align with the abovementioned initiatives, such as Griffin et al.'s [18] findings on making self-employment a viable option for persons with disabilities. They indicated that suitable and personalised environments and approaches maximise an individual's learning ability. This is particularly evident in activities such as microenterprise, where there is no conforming to conventional or traditional employment settings [21]. For instance, the microenterprise owner makes decisions, including about working hours. Despite the content mentioned above, self-employment-specific research for persons with disabilities remains remarkably scarce. Malhotra et al. (2024) [22] and Sodhi et al. (2024) [23] highlight the significant void in existing literature, while comprehensive literature reviews reveal limited publication output between 1996 and 2014 [24]. Renko et al. (2016) [25], further substantiate this gap, calling for increased research attention to self-employment for persons with disabilities. Thus, it is crucial to consider the ramifications of such employment practices by exploring the impact.

## 1.3. Impact

The above suggests that this is a complex field with various dynamics that those involved should pay attention to, including laws, context and availability of resources to ensure meaningful success and sustainability. Ultimately, the success of microenterprises impacts the lives of those involved, such as persons with disabilities and their families [26]. Among others, the success is characterised by achieving financial stability and improved quality of life. Adding on the benefits, Chimara et al. (2024) [27] suggest that initiating income-generating projects enables individuals to create and sell products, fostering autonomy and economic independence. One participant in their study emphasised the importance of self-sufficiency through entrepreneurial ventures [27]. However, achieving these outcomes requires systematic approaches, such as the involvement of others beyond the persons with disabilities, to encompass broader support networks.

## 1.4. Key role players and profession-specific involvement

Thus, gaining a nuanced understanding of microenterprises is crucial for key role players to provide adequate support and foster sustainable growth. Furthermore, self-employment necessitates a collaborative approach, where key role players from diverse backgrounds, skills, and knowledge domains work together synergistically and collaborate with the end user [18,28]. In support, a mental health study conducted in rural South Africa by Silaule et al. (2024) [29] emphasises the importance of collaborative work in such settings to benefit the end user. Additionally, achieving this goal requires the involvement of various institutions and private and community-based initiatives [18,28]. These key role players include occupational therapists who provide work or vocational services to adults, including persons with disabilities, to promote productivity and well-being [30,31]. They focus on aspects such as assessing and improving functional capacity, return to work and fostering reasonable accommodation [30,31]. Occupational therapists adopt a pragmatic and client-centred approach, putting classroom theory into practice at a clinical level [30,31]. To maximise their effectiveness and impact, it is crucial for occupational therapists to integrate specialised tools into practice.

## 1.5. Assessment and intervention

Furthermore, to enhance their role, occupational therapists can leverage microenterprise-specific tools, such as standardised forms and assessments, to add to their evaluation processes. This aligns with existing research, which supports a structured and systemic approach to self-employment in microenterprises, including utilising checklists and forms [18]. In line with the above, according to the Occupational Therapy Practice Framework: Domain and Process [31], therapists must understand individuals' employment-related interests, motivations, ability to perform work tasks, and exploration for volunteer work for exposure or capacity building. However, practical implementation requires specific application of these theoretical constructs to microenterprise contexts. Expanding on this, occupational therapists can leverage their skills, such as analysing tasks related to product making in microenterprises, to determine the associated demands and requirements. In support, Griffin et al. [18] stated the importance of evaluating one's unique skills and interests when assessing persons with disabilities' profiles for career planning or employment in microenterprises. Moreover, essential steps in establishing a microenterprise encompass identifying an individual's interests and skills related to work through interviews and observation or any other formal assessment or pencil and paper-based assessment, as well as evaluating work skills [18,32]. Subsequent steps should entail setting up the microenterprise and establishing financial resources, such as applying for relevant grants [32]. Taking it further, the International Labour Organisation (2009) [32] advocates for accreditation or issuing a certificate post-training in this type of employment. They further advocate peer training, establishing and measuring the profitability of the microenterprise, and follow-ups to evaluate how funds are used. These practical recommendations reflect theoretical principles of systematic intervention while addressing real-world implementation challenges. These imply that clinical reasoning in those facilitating such initiatives should prevail [31]. The above discussions suggest that this type of employment may be labour-intensive for parties involved, hence a systemic approach and relevant training are advisable. The necessity of a holistic and systematic approach is further reinforced by research in occupational therapy [27,28].

## 1.6. Study contribution and focus

This study forms part of a PhD project focusing on developing a framework for occupational therapists to facilitate self-employment among persons with disabilities. The research responds directly to the documented paucity of research and the absence of systematic, practical approaches for selecting and analysing suitable microenterprises for persons with disabilities. As such, the research question for this study is: *What are the procedures used to select and analyse microenterprises for persons with disabilities?* Therefore, this study aims to explore procedures used to select and analyse microenterprises for persons with disabilities. The study objectives further emphasise this integration by exploring i) the selection of a suitable microenterprise for persons with disabilities and ii) factors to consider when analysing microenterprises for successful outcomes. By focusing on procedural development and analytical approaches, this study directly contributes to bridging the identified theory-practice gap in this critical field.

## 2. Method

The study used a qualitative design engaging occupational therapists from various fields in South Africa. This design enabled the researcher to gather rich, contextual data from the perspectives of these participants. The study adhered to the Consolidated criteria for reporting qualitative research (COREQ) [33] as described below under the following subheadings: Research team and reflexivity, participant selection, data collection and analysis and ethical considerations.

### 2.1. Research team and reflexivity

The occupational therapists attended an initial online briefing session regarding this research before the scheduled interview dates, establishing a foundation for rapport. During this briefing, essential aspects covered included the researcher's background, research objectives and research guide questions.

Additionally, the occupational therapists were allocated time to pose questions to the researcher. Relevant information, such as researchers' qualifications and affiliations, is provided on this study's cover or first page, summarising the researchers' capabilities. Researchers' bias, assumptions and preconceptions were mitigated by relying exclusively on participants' data, which assisted in maintaining objectivity [33].

## 2.2. Participant selection and setting

Purposive sampling was employed, and snowball sampling was utilised to augment the sample size [34]. Occupational therapists across South Africa, from various fields of practice, were included to allow for a holistic view. These participants were contacted through telephone, email, or both during the recruitment phase. One prospective participant failed to follow through, despite initial interest, and remained uncontactable despite subsequent communication attempts. A total of seventeen occupational therapists participated in this research. The occupational therapists had to be actively involved in vocational rehabilitation and have an in-depth knowledge of various communities in South Africa. Those who did not meet the criteria were excluded.

The clinically experienced occupational therapists were all interviewed online, except for one participant, for convenience and geographical accessibility. The one participant preferred a face-to-face interview. Only the occupational therapists and the interviewer were present during the interviews. Vocational rehabilitation services in a South African context are offered mainly by occupational therapists in the private and public healthcare systems (where the Department of Health employs them at a hospital), the Department of Education (at special needs schools), or the Department of Employment and Labour offices.

## 2.3. Data collection and analysis

The recruitment and data collection ranged from 01 July 2023 to 18 January 2024. A question guide was developed, informed by literature and aligned with this study's question and objectives. Before data collection, the question guide and probing questions were piloted for feasibility and reliability. Subsequent adjustments included, but were not limited to, rephrasing and refining questions and incorporating clarifying synonyms. Such questions included: "Share with me micro-enterprises that you have come across in South Africa that you think persons with disabilities could explore... Be as specific as possible." The final question guide, key terms, and definitions were distributed to and reviewed with all participants during the briefing session and displayed on-screen during the semi-structured interviews. All interviews were digitally audio-recorded and transcribed verbatim. Data saturation was achieved, indicating that no new information emerged, and data redundancy was confirmed, ensuring thorough coverage of the research topic [34,35].

Member checking was conducted after all the occupational therapists responded to a question to verify shared understanding between the interviewer and occupational therapists [35]. No changes were made after member checking. This process was further enhanced by allowing the occupational therapists to revisit and clarify previous responses and add or modify information before terminating the interviews.

The NVIVO software was utilised to organise and analyse the data. Data was verified by reading each transcript while listening to the corresponding audio recording [33,25]. Thematic analysis was employed during the integrative hybrid inductive and deductive approach [36]. Each theme that resulted from thematic analysis is presented in the findings below with participants' quotations providing contextual support. To enhance trustworthiness, the corresponding author presented interpreted data to the other two authors for review and validation [36–38].

## 2.4. Ethical considerations

The University of KwaZulu-Natal Biomedical Research Ethics Committee (BREC) issued ethical clearance (No. BREC/00004655/2022). Furthermore, permission was obtained from relevant gatekeepers, and participants were fully informed through information sheets and provided signed informed consent.

## 3. Findings

In this section, the participants' demographics will be described, as well as the two themes that emerged.

### 3.1. Participants' demographics

Participants in this research were occupational therapists based at Learners with Special Education Needs (LSEN) Schools, institutions of higher learning, and clinically based settings. Refer to Table 1 for details. All participants (n = 17) were occupational therapy service providers:

• LSEN Schools (n = 5)

• Institution of higher learning (n = 6)

• Clinically based (n = 6)

Their average years of experience in occupational therapy in private and public settings were 14 years. They practised and registered with their professional body, the Health Professions Council of South Africa (HPCSA). These participants possessed vocational rehabilitation expertise and familiarity with urban and rural settings across South Africa.

Although occupational therapists from the Department of Employment and Labour participated, their input and approach were unsuitable for this study. The mandate of the Department of Employment and Labour is to regulate the South African labour market for sustainable economic development.

### 3.2. Themes

Table 2 and the paragraphs below outline the themes and subthemes. Direct quotes represent the participants' voices after each theme or subtheme.

**Table 1. Demographics of occupational therapists.**

| Current setting | | Pseudonym | Gender | Qualification<br>Bachelor (B); Bachelor of Science (BSc); Master (M); Master in Public Health (MPH); Master of Science (MSc); Occupational Therapy (OT); Vocational Rehabilitation (VR) | Years of practice |
|---|---|---|---|---|---|
| Learners with Special Education Needs School | 1. | JR | Female | B OT | 21 |
| | 2. | Lune | Female | B OT | 21 |
| | 3. | AP | Female | B OT | 31 |
| | 4. | Blue Flower | Female | B OT | 15 |
| | 5. | Taxido Cat | Female | B OT | 25 |
| Institutions of higher learning | 6. | Mac | Male | BSc OT, Postgraduate Diploma in VR & M OT | 10 |
| | 7. | Participant | Female | B OT & M OT | 10 |
| | 8. | Shera | Female | BSc OT & MPH | 21 |
| | 9. | Blue | Female | BSc OT & MPH | 12 |
| | 10. | Participant 05 | Female | B OT & MSc Inclusive Environments | 23 |
| | 11. | Participant 00 | Male | BSc OT & MSc OT | 9 |
| Clinical based | 12. | Number 5 | Female | B OT & Postgraduate Diploma VR | 7 |
| | 13. | Butterfly | Female | B OT & M OT | 3 |
| | 14. | Emerald | Female | B OT, Postgraduate Diploma VR & MOT | 5 |
| | 15. | Brown | Male | B OT & Diploma OT | 21 |
| | 16. | Arthur | Female | B OT & MPH | 6 |
| | 17. | Hedgehog | Female | BSc OT & MSc OT | 5 |

**Table 2. Overview of the themes and subthemes.**

| Theme | Subtheme | |
|---|---|---|
| 3.2.1. Theme one: The process of selecting an appropriate microenterprise | 3.2.1.1. | Approaches to microenterprise selection |
| | 3.2.1.2. | A list of microenterprises found in a South African context for career choice consideration |
| | 3.2.1.3. | Considerations when selecting microenterprise subcategories |
| 3.2.2. Theme two: Factors to consider when analysing a suitable microenterprise as a placement option | 3.2.2.1. | Microenterprise accessibility |
| | 3.2.2.2. | Involvement of various key role players |
| | 3.2.2.3. | Physical, cognitive and mental demand factors |
| | 3.2.2.4. | Funding availability and access factors |

**3.2.1. Theme one: The process of selecting an appropriate microenterprise.** Theme one focuses on the significance of a comprehensive approach to selecting a suitable microenterprise for self-employment for persons with disabilities, according to occupational therapists. The pivotal aspects they highlighted are the approach to microenterprise selection, the microenterprise list or options available and the distinct features of these microenterprises. The occupational therapists stressed that these interrelated factors are crucial when determining an appropriate microenterprise that aligns with the interests, skills, and goals of persons with disabilities exploring self-employment.

3.2.1.1. Approach to microenterprise selection: To ensure efficiency and optimal outcomes, occupational therapists emphasised that selecting a suitable microenterprise for persons with disabilities requires a systematic and practical approach. The following are direct quotes from the occupational therapists:

*"There are systems that need to be put in place before you start* [selecting a suitable microenterprise] *with the patient."* (Blue)

*"We* [occupational therapists] *need to take a practical step into what's achievable for the individual."* (Number 5)

3.2.1.2. A list of microenterprises found in a South African context for career choice consideration: The occupational therapists provided a list of microenterprises typically found in a South African context for consideration, exploration or use when selecting a suitable microenterprise for persons with disabilities. Refer to Table 3 for a complete list of microenterprises. They indicated that persons with disabilities can select microenterprises with support from an occupational therapist, which can occur during initial therapy stages, screening, or at the point of entry, e.g., at special education settings or community healthcare centres. As Mac reported, *"Start* [therapy] *from the first point of screening."* The occupational therapists further reported that formal tests, a guide, or a checklist would help establish the suitability of a job (microenterprise). In their view, the microenterprises list will facilitate career choices, set goals, and provide direction towards employment for persons with disabilities. Below are some supporting quotes:

*"[We need some]* *kind of guidelines, like almost like a checklist."* (Number 5)

*"You need to assist them* [persons with disabilities] *in terms of the direction and the goal setting."* (Butterfly)

*"Assessing which learners* [or persons with disabilities] *would be suitable for which sort of jobs, for example* [using] *the T/PAL test* [Therapists' Portable Assessment Lab], *you know, it gives you guidance in terms of career choices."* (JR)

3.2.1.3. Considerations when selecting microenterprise subcategories: The list of microenterprises shared by occupational therapists had diverse features, and they emphasised that the list comprises contextually relevant and day-to-day microenterprises commonly found in a typical South African context. Refer to Table 3 for a complete list of microenterprises and to Table 4 for associated subcategories and factors. Some notable features they reported are that these

**Table 3. Categorised list of microenterprises commonly found in South African contexts.**

| Retail (Buy & sell products) [6],[39] | Service/skills (Offer service or use skills) [6],[39] | Manufacturing/production (Making, producing and growing products) [6],[39] |
|---|---|---|
| **[BEAUTY]**<br>1. Hair weaves and hair products<br>2. **[CLOTHING/APPAREL]**<br>3. Clothing (e.g., "traditional-wear – shwe shwe fabric")<br>4. Underwear<br>5. Shoes<br>6. Handbags<br>7. **[FOOD] - raw-cooked/street**<br>8. [Cold beverage] Ginger beer and [soft drinks and water]<br>9. [Cooked Meat] Shisa nyama [braai meat], chicken feet, Sepatlo, magwinyas [fat cakes], boiled eggs, fried chips, pap and mogodu<br>10. [Hot beverage at, e.g.,] Coffee barrister or café<br>11. Coffee/Tea & [condiments], e.g., bread, muffins/fat cakes, samosa<br>12. [Uncooked Meat] Chicken, braai packs or packaged meat, e.g., in rural areas; eggs<br>13. Fruits & Vegetables<br>14. Restaurant<br>15. Snacks (sweets, biscuits, chips and nuts)<br>16. **[HOUSE WARE]**<br>17. Buckets and basins (for bathing and cleaning)<br>18. Cleaning materials [e.g., for home]<br>19. Florist<br>20. Gardening services<br>21. Sand [e.g., for building]<br>22. Selling cosmetics/toiletries<br>23. Sponges, toothbrushes, face towels, etc<br>24. Tapper ware<br>25. Umbrellas<br>26. Wheelbarrow<br>27. Wood [e.g., for fire, house roofing]<br>28. **[MISCELLANEOUS/OTHER]**<br>29. [Education] Books, stationery, past [exam] papers<br>30. Airtime<br>31. Brand ambassadors, e.g., joining chain store companies such as Avon and Herbalife<br>32. Bricks<br>33. Cars (used)<br>34. Cell phones (used), covers and screen protectors<br>35. Cigarettes<br>36. Electrician<br>37. Laptops<br>38. Promotional/ food items during community events<br>39. Spaza shop or tuckshop [e.g.,] in an industrial area<br>40. Tyre business (second hand) | **[BEAUTY]**<br>1. Beauty Therapy and Salons [offer services such as:]<br> • Hairdressing [e.g.,] hair styling and haircuts<br> • Make-up<br> • Massage<br> • [Nails] Manicure and pedicure<br>2. **[CATERING, ENTERTAINMENT & EVENTS]**<br>3. Catering<br>4. Hospitality<br>5. Hiring or renting out<br> • Mobile fridges or toilets [e.g.,] during community events<br> • Catering equipment, e.g., tables, chairs, cutlery and crockery<br> • Sound equipment, e.g., a speaker and mic system for weddings or funerals<br> • **[TECHNOLOGY]**<br> • Basic IT [Information technology] services<br> • Computer services [and] Internet café, e.g., internet access, drawing up CVs, online research for others (including online booking for home affairs, and online registration for voting)<br> • Content creation, [e.g.,] physical disability tips for social<br> • Document photocopy and printing services, e.g., identity documents (IDs) and curriculum vitae (CVs)<br> • Graphic design work, e.g., advert and pamphlet design, such as programmes for memorials within the community<br> • Photography and video[graphy], e.g., picture framing, taking ID photos, family pictures, and school pictures (printed and digital pictures)<br> • **[DOMESTIC]**<br> • Babysitting service, e.g., school aftercare service (supervision and provision of food)<br> • Cleaning services (industrial & domestic or household), e.g., laundry, ironing and window washing<br> • Garden services, e.g., flower arrangement, grass cutting, landscaping and tree felling<br> • Grocery service, e.g., shopping and delivery<br> • Homework assistance or housekeeping and basic [household] maintenance [general worker], e.g., changing a light bulb<br> • Sneaker wash or shoe cleaning<br> • **[SCHOOL/ACADEMIC]**<br> • Book covering services<br> • Extra[curricula] lessons, e.g., music instruments and swimming lessons<br> • Lecturing<br> • [Tuition] provision of school tuition<br> • Tutoring, e.g., [subject] and online language teaching | **[CLOTHING & FASHION]**<br>1. Bags<br>2. Beading, e.g., making traditional jewellery<br>3. Earrings making<br>4. Sewing [with given patterns] clothing or dressmaking, e.g., Traditional attires using shweshwe [fabric] or sleepwear<br>5. **[FOOD]**<br>6. Baking, e.g., muffins, bread and cakes for weddings & birthdays<br>7. Food production or cooking<br>8. **[FARMING]**<br>9. Breading, e.g., hunting dogs<br>10. Crop, e.g., vegetables, herbs and hot plants<br>11. Egg farming or production<br>12. Livestock farming, e.g., poultry (chicken and ostrich) and piggery<br>13. **[DOMESTIC & HOUSEWARE]**<br>14. Bedding<br>15. Blankets, e.g., patched blankets<br>16. Broom making, e.g., from straw and grass-weaved brooms<br>17. Candle making<br>18. Ceramics<br>19. Cleaning material manufacturing, e.g., soap and floor polish<br>20. Décor items<br>21. Kilt [making] from material offcuts<br>22. Macrame, e.g., making wall or pot hangers<br>23. Making and selling sewn products such as ironing board covers, aprons, peg bags, etc<br>24. Mats making, e.g., floor mats [made from] grass<br>25. Pottery, e.g., pot planters<br>26. Sewing, e.g., custom-made curtains, school tracksuits and church uniforms<br>27. Toilet [paper and] roll manufacturing<br>28. **[OTHER]**<br>29. [Clay work], e.g., polymer clay<br>30. Book covering<br>31. Bricks or blocks making<br>32. Candle making<br>33. Crochet – making big mats<br>34. Fence making<br>35. Hammock<br>36. Hand-made stationery, pencils, rulers etc.<br>37. Knitting<br>38. Leather products, e.g., wallets and whip<br>39. Low-cost products, e.g., toys and low-cost sanitary pads<br>40. Masonry<br>41. Pillows making |

*(Continued)*

**Table 3.** (Continued)

| Retail<br>(Buy & sell products) [6],[39] | Service/skills<br>(Offer service or use skills) [6],[39] | Manufacturing/production<br>(Making, producing and growing products) [6],[39] |
|---|---|---|
| 1. **[REPAIR]**<br>2. Cell phones repairs<br>3. Computer, e.g., laptop repair<br>4. Electronics [appliances], e.g., kettle and irons<br>5. Sewing [e.g., mending torn clothes]<br>6. Sound systems<br>7. Tyre repairs<br>8. Wheelchair repair<br>9. **[COBBLER]**<br>10. Shoe repairs<br>11. Polishing<br>12. **[TRADE-ARTISAN]**<br>13. Artistic skills<br>14. Bricklaying<br>15. Painting<br>16. Plumbing<br>17. Security bars fitting<br>18. Upholstery services<br>19. Welding<br>20. [Window] services, e.g., blinds fitting<br>21. **[OTHER]**<br>22. Administration work, e.g., office administration, working a printer<br>23. Auxiliary health workers<br>24. Bible school<br>25. Car wash services<br>26. Car washing<br>27. Cleaning services<br>28. Décor<br>29. Deliveries, e.g., goods<br>30. Packaging and categorisation<br>31. Transportation – e.g., school transport services | 1. Recycled material products<br> • Cattle feeding basins from tyres<br> • Chairs<br> • crotchets mats<br> • Floor ports<br> • Holder (bulb or candle holder) from glass bottles<br> • Sun hats from plastic bags<br> • table cloths from plastic bags<br>1. Shoes making<br>2. Tablecloths making<br>3. Trays, e.g., for serving people<br>4. Art, craft and technical work<br> • 3D printer<br> • Card-making<br> • Carpentry<br> • Keyrings<br> • Laser cutter<br> • Furniture making<br> • Metalwork, e.g., steel structures and carports<br> • Ornaments<br> • Sculptures<br> • Stepping stools<br> • Trinkets<br> • Woodwork | |

microenterprises can be operated by individuals, families or groups, depending on their complexity, level of operation or position within the business value chain, i.e., ranging from production to sales of various items. These microenterprises encompass subcategories and associated factors, as highlighted in Table 3 below.

**3.2.2. Theme two: Factors to consider when analysing a suitable microenterprise as a placement option.** The occupational therapists highlighted factors that need analysis after selecting a suitable microenterprise for persons with disabilities. To provide context, Emerald noted,

> "*We* [occupational therapists] *really, really need to break down these things* [self-employment in microenterprises] *and look at the realities of our low-income areas and the disability issues that come with them.*"

Specifically, the occupational therapists reported that the analysis should encompass the microenterprise accessibility, the key role players involved, business demands, and funding availability and access. They believe occupational therapists can facilitate successful job placement in microenterprises by understanding these interconnected factors.

**Table 4. Microenterprises subcategories and associated factors.**

| Subcategories | Associated factors |
|---|---|
| • Beauty<br>• Apparel, clothing & fashion<br>• Domestic or household goods<br>• Events and entertainment<br>• Farming<br>• Food and catering<br>• Repair services<br>• School/academic support<br>• Technology<br>• Trade-artisan | • Areas of operation – Peri-urban, urban and rural<br>• Manual or technology use, e.g., hand washing and machine washing<br>• Sizes vary, ranging from basic to sophisticated, e.g., the level of customer service or the number of branches the microenterprise has<br>• Mobile, stationary, fixed or offer delivery<br>• New, thrift, used, second-hand, or pre-owned items<br>• Operate daily or occasionally<br>• Some are combined or are stand-alone or at an existing establishment, e.g., a coffee barrister in a convenience store<br>• Store, side of the road, industrial area, or pop-up |

3.2.2.1. Microenterprise accessibility: The occupational therapists reported that a microenterprise can take various forms, including contact-based or online business models, which are informed by the selected suitable business category (retail, service-oriented or manufacturing). Notably, some microenterprises may require transportation to obtain stock or access customers. Furthermore, they stated that multiple factors must be considered, particularly environmental challenges, such as accessible roads, where access presents a significant obstacle, especially for persons with disabilities who rely on assistive devices such as a wheelchair. The occupational therapists had the following to say:

*"It could be in a physically inclined environment or whatever the person is looking into locating themselves. It could be an online type of thing."* (Mac)

*"[Consider] infrastructure and accessibility. If you look at clients in low-income areas, if they need to buy stock and things like, you know, accessibility like the roads, if they are in poor condition, people in wheelchairs [will have difficulties]... How do you expect them to go and buy stock and come back at a reasonable cost?"* (Emerald)

*"…Transport is [a] huge cost."* (Arthur)

*"There is a lot of limitations in our communities in terms of mobilising around and lots of environmental barriers… Environmental barriers are massive"* (Hedgehog)

3.2.2.2. Involvement of various key role players: The occupational therapists stressed the importance of understanding, referring further and working with other key role players involved in microenterprises. Their perspective is that collaboration between persons with disabilities and involved key role players will contribute to establishing and sustaining the microenterprise. The occupational therapists reported that these key role players could include family members, multidisciplinary professionals, community leaders, and business owners who can teach business skills. Below are some supporting quotes:

*"Leave the responsibility [of running the microenterprise] with the service user [persons with disabilities]."* (Participant)

*"Also, [consider] identifying somebody who can provide support outside [the hosting institution], you know, like a family member."* (Taxido Cat)

*"I think what the OT [occupational therapist] knows may be sufficient to help that person to start something small... There are these pockets of support, like informal support structures [that should be considered]."* (Mac)

*"They [persons with disabilities] should be putting themselves out in a community level, whether it's with the community leader, whether it's, it's prominent people in the community, such as traditional healers, like those sort of people like, they need to make themselves known to those leaders to, to have support."* (Emerald)

*"In some of the skills where occupational therapists are limited, they should use the multidisciplinary team… Say we're looking at self-employment for people with mental health issues, then we know that we will need a doctor, we will need like a psychiatrist, we'll need a psychologist."* (Participant 00)

*"But also part of that, like getting other business people involved so that they teach us* [occupational therapists]*, including the people with disabilities, some* [business] *skills."* (Participant 00)

3.2.2.3. Physical, cognitive and mental demand factors: The occupational therapists indicated that persons with disabilities should meet the microenterprise demands, which typically involve physically handling tasks and possessing sufficient cognitive and mental abilities to manage business operations. Also, demonstrating resilience, adaptability and a strong commitment to overcome various business challenges and obstacles were reported as essential. Here are some supporting quotes:

*"Some* [persons with disabilities] *can do the* [physical] *job, but they don't have that level of reading. They need to have a certain level of reading."* (Blue Flour)

*"Being self-employed is not a, a child's play. Regardless of how big or small it is, the person must commit."* (Mac)

*"Motivation and resilience are the two things that are enablers in terms of facilitating the self-employment process."* (Arthur)

*"Their* [persons with disabilities] *own volition, and motivation is an enabler."* (JR)

3.2.2.4. Funding availability and accessibility factors: The occupational therapists highlighted how crucial capital or start-up funding is in establishing and sustaining a microenterprise. They suggested several possible funding sources, such as from persons with disabilities (personal savings, disability grant fund, or lump sum payouts from insurance policies) or external sources such as family, friends or funding institutions. The occupational therapists had the following to say:

*"A big part of self-employment is capital."* (Hedgehog)

*"A start-up funding often is what deters most people to start."* (Shera)

*"[Persons with disabilities could use] their* [disability] *grants, and then they start up a little business."* (Blue Flour)

*"If they are an injury on duty* [case]*, so workman's comp.* [compensation]... *Those that are insured with group risk and individual policies... And even disability income...* [These avenues can be used as capital injection since business] *capital is a problem."* (Arthur)

## 4. Discussion

The findings of this study reveal critical insights into how occupational therapists can systematically facilitate microenterprise selection and placement for persons with disabilities in the South African context. The two primary themes that emerged, namely the process of selecting appropriate microenterprises and factors for analysing suitable placement options, demonstrate the complexity and multi-layered nature of vocational rehabilitation in self-employment contexts. This discussion examines how these findings bridge the gap between theoretical frameworks and practical implementation in occupational therapy practice.

### 4.1. Selecting suitable microenterprises

The study's findings demonstrate that occupational therapists recognise the need for systematic, collaborative approaches when facilitating microenterprise selection for persons with disabilities. This aligns fundamentally with established

occupational therapy principles of client-centred practice and comprehensive assessment protocols. Aligning with this research is the profiling of an individual in occupational therapy, which involves, but is not limited to, assessing their function, goals and plans to ensure tailored interventions to meet their needs [31]. This systematic process utilises various evaluation methods, including interviews, observations of task performance, standardised tests and completion of relevant assessment forms [31]. The participants' emphasis on "systems that need to be put in place" and taking "a practical step into what's achievable for the individual" reflects this established profiling approach. However, it should be cautioned that sound clinical reasoning is crucial for the therapist to offer tailored, case-specific services and adapt when the need arises, i.e., no two individuals are alike [31].

The comprehensive microenterprise list (Table 3) generated from this study represents a significant contribution to practice, providing occupational therapists with contextually relevant options spanning retail, service, and manufacturing sectors. The list reflects microenterprises from urban to rural South African settings [2,5,6,40]. The same is true for microenterprises in other parts of the world, i.e., they have similarities with those found in South Africa [1,3,4,7–17]. This finding addresses a critical gap in vocational rehabilitation resources, as participants specifically called for "guidelines, like almost like a checklist," and formal assessment tools. The categorisation into beauty, clothing, food, domestic services, technology, and trade-artisan subcategories provides practitioners with structured frameworks for matching individual capabilities with realistic employment opportunities.

Importantly, the study reveals that these South African-based microenterprises' features were presented as relatively simple, primarily informal or unregistered, requiring minimal start-up capital, and are small-scale or solo ownership ventures. Some have the potential for scalability [40,41]. These microenterprises are often based at either the owner's home or a business site, demanding a combination of manual, physical and/or cognitive abilities. Ownership encompasses a range of skill levels, from skilled to unskilled. Notably, some leverage on technology and monetary benefits, for instance, accrue immediately for some microenterprise owners [40]. However, the findings also highlight that these dynamics may deter some, while others may lack the capacity. These highlight the intricacies and nuances and emphasise the fact that not all persons with disabilities can take part in self-employment, requiring careful assessment and realistic goal-setting approaches.

The microenterprises described above are categorised into three primary types: retail (buying and selling), service-oriented, involving the utilisation of specific skill sets, and manufacturing/production, which entails producing, making and selling a product [39,42,43]. To enhance the practicality and realism of microenterprise selection, the microenterprises list in Table 3 could be reorganised or formatted to incorporate time-based parameters: "past", "present", and "future". Refer to Table 5. This framework would enable persons with disabilities, while being assessed by an occupational therapist, to utilise these dimensions for each microenterprise (e.g., running a shop) to indicate whether they have previously engaged in the activity ("past"), are currently engaged ("present") or aspire to engage in it ("future").

### 4.2. Factors to consider when analysing a suitable microenterprise as a placement option

The key role player collaborating with the end user (persons with disabilities) should understand contextual complexities [29] and the implications of establishing and maintaining a microenterprise. The participants in this study support this perspective. The second theme reveals the multifaceted analysis required after initial microenterprise selection, emphasising four critical domains: accessibility, key role player involvement, demand factors, and funding considerations. This encompasses understanding the legalities, including whether registration for the microenterprise is mandatory. Moreover, understanding associated regulatory requirements, as observed in a study by Monareng et al. (2024) [41], who noted that these businesses often operate informally and lack formal registration, and as a result, do not have access to traditional commercial banking services (36). However, it should be noted that the above or accessibility matters do not exempt persons with disabilities from having the necessary knowledge about this field, especially since a collaborative approach is emphasised.

**Table 5. Categorised list of possible self-employment in microenterprises with time-based parameters [Insert microenterprise name under 1-4 and add microenterprise rows as required].**

| | | Past | | | | | Present | | | | | Future | | | | |
|---|---|---|---|---|---|---|---|---|---|---|---|---|---|---|---|---|
| | | **Retail** | | | | | | | | | | | | | | |
| **Rating** | | 1 | 2 | 3 | 4 | 5 | 1 | 2 | 3 | 4 | 5 | 1 | 2 | 3 | 4 | 5 |
| 1 | | | | | | | | | | | | | | | | |
| 2 | | | | | | | | | | | | | | | | |
| 3 | | | | | | | | | | | | | | | | |
| 4 | | | | | | | | | | | | | | | | |
| | | **Service/skills** | | | | | | | | | | | | | | |
| **Rating** | | 1 | 2 | 3 | 4 | 5 | 1 | 2 | 3 | 4 | 5 | 1 | 2 | 3 | 4 | 5 |
| 1 | | | | | | | | | | | | | | | | |
| 2 | | | | | | | | | | | | | | | | |
| 3 | | | | | | | | | | | | | | | | |
| 4 | | | | | | | | | | | | | | | | |
| | | **Manufacturing/production** | | | | | | | | | | | | | | |
| **Rating** | | 1 | 2 | 3 | 4 | 5 | 1 | 2 | 3 | 4 | 5 | 1 | 2 | 3 | 4 | 5 |
| 1 | | | | | | | | | | | | | | | | |
| 2 | | | | | | | | | | | | | | | | |
| 3 | | | | | | | | | | | | | | | | |
| 4 | | | | | | | | | | | | | | | | |
| **Subtotal** | | | | | | | | | | | | | | | | |
| **Grand Overall total** | | | | | | | | | | | | | | | | |
| Summarise or interpret ratings in collaboration with PWDs(High scores may suggest possible placement options) | | | | | | | | | | | | | | | | |

Building on the above, Griffin et al. (2003) [18] emphasise the importance of equipping key role players, such as rehabilitation professionals, with tools and resources to facilitate microenterprise development, including necessary guidelines, steps, and forms. This is essentially the finding in this research, where the occupational therapists, reported that working in collaboration with other key role players and conducting a thorough analysis of the selected microenterprise's nature is necessary to determine what support structures need to be in place, including further training and funding requirements. Specifically, this involves identifying potential funding sources and strategies for accessing business finances and understanding the procedures for advocated by participants obtaining business funds [32]. On upskilling, Griffin et al. (2003) [18] suggested that consideration should be given to peer training facilitated by community structures or local experts, such as local business initiatives, village leaders, and self-help groups. They added that business-related training opportunities could focus on stock management (e.g., sourcing and inventory management) and budgeting. Again, the above highlights the potential complexities or dynamics that are to be taken into account at all times by both the therapist and the individual receiving the service.

Moreover, utilising effective problem-solving strategies is crucial to address challenges related to customer base and access [18], a view supported by occupational therapists in this research. For instance, considering alternative business sites, such as opting for an online instead of a home-based enterprise, can mitigate physical access challenges for remote areas. This strategy can be successfully executed in collaboration with experts or those with expertise in concerned

communities, i.e., competent individuals [18,41]. Further attention should be paid to the resources available to persons with disabilities, including support from family and friends [41] and their internal strengths, such as physical and cognitive abilities. According to the participants, tailored interventions or programs should incorporate these elements where necessary, as they have the potential to mitigate complexities associated with self-employment and make it successful and sustainable. The suggestions above closely align with the expertise of occupational therapists, particularly in vocational rehabilitation, and their referral aspect for further support [27,28,31].

According to one of the participants, Blue, the effective running of a microenterprise requires comprehensive knowledge and skills, which involve building abilities to ensure successful business operations. In support, the International Labour Organisation (2009) [32] suggested further forms related to microenterprises, such as training completion forms (peer training), business start-up and enhancement, measuring profitability, application for grants or loans, evaluation of the participant's use of funds and business follow-up forms. These findings are further supported by Griffin et al. (2003) [18], who proposed supplementary forms with aspects such as i) skills and needed supports, and ii) a business refinements chart. Such forms play a critical facilitative role in systematically approaching self-employment in microenterprises. The skills and needed supports aspect comprises three essential components: I can do this, I need these supports, and Who can help – the latter referring to a personal contact or a professional to be hired, such as an accountant. Such forms are vital in assessing various business functions, including inventory management, marketing strategies, and sales operations [18]. Furthermore, the *Business Refinements Chart* is a valuable tool in refining the microenterprises' core business elements, including company name, targeted customers, business uniqueness, services and products offered, and why customers should buy their products [18]. The above align with Tables 3 and 5, respectively. Although necessary, this may be labour-intensive for the collaborators, so services and tasks should be tailored and offered as needed and at the parties' discretion.

Given the widespread presence of microenterprises, these factors or considerations collectively should foster a sustainable and competitive advantage [1–17]. The global prevalence of microenterprises, requires key role players to leverage opportunities to empower persons with disabilities, i.e., make self-employment a viable job option. Ultimately, strategic microenterprise selection and analysis, effective matching and placement are critical avenues that occupational therapists should explore. However, as discussed above, self-employment in microenterprises may not be as straightforward given the various dynamics in this field. Overall, an occupational therapist should use their clinical reasoning, strive to select appropriate and relevant assessments for vocational evaluation considering the skills required to perform a job (microenterprise) and the demands a job places on a worker [44].

### 4.3. Implications for theory-practice integration

These findings collectively demonstrate that microenterprise-focused vocational rehabilitation requires occupational therapists to integrate traditional clinical skills with entrepreneurship support competencies. The systematic approach advocated for by participants – from initial screening through ongoing business support – mirrors occupational therapy's commitment to comprehensive, client-centred intervention while expanding into previously unexplored practice domains.

The study reveals that successful microenterprise placement requires occupational therapists to function as vocational counsellors, community liaisons, business consultants, and accessibility advocates simultaneously. This expanded role definition has significant implications for professional development, suggesting the need for specialised training programs that integrate vocational rehabilitation principles with entrepreneurship support methodologies.

Furthermore, the findings indicate that microenterprise development represents a promising avenue for addressing unemployment among persons with disabilities in resource-constrained contexts, provided that occupational therapists develop appropriate assessment tools, collaborative networks, and support frameworks. The comprehensive nature of factors identified – from accessibility considerations to funding strategies – suggests that successful implementation requires systematic approach development rather than ad-hoc interventions.

 

The study's emphasis on contextual relevance and community-based solutions aligns with contemporary occupational therapy movements toward community-centred practice while addressing the urgent need for innovative employment solutions for persons with disabilities in developing contexts. However, as participants noted, the complexity of microenterprise development means that "not all persons with disabilities can take part in self-employment", requiring careful assessment and realistic goal-setting approaches that honour individual capacities and limitations while maximising potential for occupational participation and economic empowerment.

## 5. Conclusion and recommendations

This study explored microenterprise selection and suitability analysis to ensure success for persons with disabilities. The findings emphasise the importance of a comprehensive and systematic approach to facilitate self-employment as a viable career choice. Among the proposed approaches to enhance microenterprise selection was exploring contextual microenterprises and using forms with time-based parameters ("past," "present," and "future") to establish persons with disabilities' profiles and explore their interests. This research identified distinctive key features of microenterprises, including their informal nature and minimal start-up capital requirements.

Other essential factors to consider when evaluating microenterprises include contact-based or online business models, legal and regulatory frameworks, funding sources, collaborative efforts among those involved, leveraging the strengths of persons with disabilities, and tailored training needs. These factors, effectively integrated with occupational therapists' expertise in vocational rehabilitation, can enhance the vocational success of persons with disabilities.

However, contemporary research on self-employment among individuals with disabilities continues to be lacking, necessitating further research to address this knowledge gap. As such, future studies should prioritise collecting more context-relevant data and collaborating with key role players to inform evidence-based practices and policies. Thus, the recommendations from this research are summarised as follows:

- Utilise a small sample size to pilot and refine the suggested systematic approach, and validate its effectiveness before considering large-scale implementation.

- Collaborate with key role players to identify their critical roles and responsibilities in promoting self-employment among occupational therapists.

- Conduct longitudinal studies beyond the South African context to inform the development of comprehensive self-employment resources for occupational therapists.

- Identify pertinent policy gaps, barriers, and facilitators to inform evidence-based recommendations.

Furthermore, it is recommended that persons with disabilities should rate their interest in the microenterprise (e.g., running a shop) using a standardised scale system such as a 5-point Likert scale. Refer to Table 5 below. The ratings for all listed microenterprises will be aggregated and analysed upon completion of this process. Higher ratings would indicate a greater likelihood of interest in a specific microenterprise (e.g., running a shop) or a microenterprise category (e.g., retail or manufacturing), potentially informing vocational placement recommendations or career choice. As a way forward, analysing the selected or microenterprise of choice would be necessary to assess their operational dynamics.

## Acknowledgments

The authors acknowledge the participants for sharing their expertise and advancing knowledge on self-employment in microenterprises for persons with disabilities.

## Author contributions

**Conceptualization:** Luther Lebogang Monareng.

**Investigation:** Luther Lebogang Monareng.

**Supervision:** Shaheed Mogammad Soeker, Deshini Naidoo.

**Writing – original draft:** Luther Lebogang Monareng.

**Writing – review & editing:** Shaheed Mogammad Soeker, Deshini Naidoo.

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
