## [Decision Letter · Decision Letter 0]

9 Jul 2025

Dear Dr. Monareng,

Thank you for submitting your manuscript to PLOS ONE. After careful consideration, we feel that it has merit but does not fully meet PLOS ONE’s publication criteria as it currently stands. Therefore, we invite you to submit a revised version of the manuscript that addresses the points raised during the review process.

We look forward to receiving your revised manuscript.

Kind regards,

Ali Junaid Khan, PhD

Academic Editor

PLOS ONE

Journal Requirements:

3. In the online submission form, you indicated that data can be requested by contacting the main author.

4. Please remove all personal information, ensure that the data shared are in accordance with participant consent, and re-upload a fully anonymized data set.

Reviewers' comments:

Reviewer's Responses to Questions

**Comments to the Author**

1. Is the manuscript technically sound, and do the data support the conclusions?

Reviewer #1: Partly

Reviewer #2: Partly

2. Has the statistical analysis been performed appropriately and rigorously?

Reviewer #1: No

Reviewer #2: N/A

3. Have the authors made all data underlying the findings in their manuscript fully available?

Reviewer #1: No

Reviewer #2: Yes

4. Is the manuscript presented in an intelligible fashion and written in standard English?

Reviewer #1: No

Reviewer #2: No

Reviewer #1: The “Introduction” part consists of theoretical discussion. There should be a separate “Literature Review” Part. The paper should demonstrate an adequate understanding of the relevant latest literature in the field and cite an appropriate range of literature sources. The “Methods” part is not well-written. The methods are not employed appropriately and able to meet the objectives of the study. The “Results” part is not well-written. The conclusion does not adequately tie together the other elements of the paper.

Reviewer #2: I have listed some constructive comments to criticize the weaknesses of the paper as below.

Constructive Comments:

1. At first glance, the aim, objectives, and methodology of the paper seem a bit vague. There are little adequate clarifications. I recommend to the authors to specify the aim of the study in the abstract section more appropriately.

2. In the introduction section, the authors can reflect the relevant literature review in a more systematic way. In the current form, there is a mess in the paper. In addition, the questions can be listed in the introduction to attract the attention of the readers.

3. I recommend to the authors to clarify their methodology in detail, making sure that their planned methods/research tools are fully detailed. Some examples from the interview questions can be added in this section. The authors ought to give attention to justifying the chosen methodology in terms of demonstrating applicability, adjustment, and usefulness in the paper.

4. Theory-practice nexus is a good tool for a “thesis vs. counter-thesis = synthesis” approach. In this context, the way how the authors support and proof their argument and falsify their counterarguments can be more effective to keep up being focused on their story. So that the central research questions and the “argument vs. counter-arguments interactions” are streamlined in a way that addresses the research inquiries in a systematic and effective manner. Some references regarding stakeholder theory can be adapted in the theoretical framework. Thus the authors can integrate the discussion part with conclusion and recommendation section.

7. In fact, I appreciate there has been a lot of reading and ground covered, but this will not appeal to readers if the paper lacks a strong focus, compelling argument and discussion, and an indication of why the paper holds value to the readership of PLOS ONE.

8. Finally, the investigation does not attain a theoretical saturation which comprises universally academic validity, reliability, clarity, intelligibility, and depth.

Some Technical Critiques and Recommendations for the Author:

1) The author should strictly adhere to submission guidelines of the journal.

2) There are several grammatical errors and instances of badly worded/constructed sentences. Please check the manuscript and refine the language carefully. I suggest using proof reading papers before submission.

**Do you want your identity to be public for this peer review?** For information about this choice, including consent withdrawal, please see our Privacy Policy

Reviewer #1: **Yes: ** Sayed Farrukh Ahmed

Reviewer #2: **Yes: ** Dorian Aliu

---

## [Author Response · Author response to Decision Letter 1]

18 Aug 2025

Kindly request that the reviewers provide explicit and constructive feedback, preferably directly in the manuscript.

---

## [Decision Letter · Decision Letter 1]

28 Aug 2025

A study exploring procedures used to select and analyse microenterprises for persons with disabilities

PONE-D-25-18451R1

Dear Monareng,

We’re pleased to inform you that your manuscript has been judged scientifically suitable for publication and will be formally accepted for publication once it meets all outstanding technical requirements.

Kind regards,

Ali Junaid Khan, PhD

Academic Editor

PLOS ONE

Additional Editor Comments (optional):

Reviewer #1:

Reviewer #2:

Reviewers' comments:

Reviewer's Responses to Questions

**Comments to the Author**

Reviewer #1: All comments have been addressed

Reviewer #2: All comments have been addressed

2. Is the manuscript technically sound, and do the data support the conclusions?

Reviewer #1: Yes

Reviewer #2: Yes

3. Has the statistical analysis been performed appropriately and rigorously?

Reviewer #1: Yes

Reviewer #2: N/A

4. Have the authors made all data underlying the findings in their manuscript fully available?

Reviewer #1: Yes

Reviewer #2: Yes

5. Is the manuscript presented in an intelligible fashion and written in standard English?

Reviewer #1: Yes

Reviewer #2: Yes

Reviewer #1: (No Response)

Reviewer #2: (No Response)

**Do you want your identity to be public for this peer review?** For information about this choice, including consent withdrawal, please see our Privacy Policy

Reviewer #1: **Yes: ** Sayed Farrukh Ahmed

Reviewer #2: **Yes: ** Dorian Aliu

---

## [Editor Report · Acceptance letter]

PONE-D-25-18451R1

PLOS ONE

Dear Dr. Monareng,

I'm pleased to inform you that your manuscript has been deemed suitable for publication in PLOS ONE. Congratulations! Your manuscript is now being handed over to our production team.

Kind regards,

on behalf of

Dr Ali Junaid Khan

Academic Editor

PLOS ONE